# Comparison of the Dinoprostone Vaginal Insert and Dinoprostone Tablet for the Induction of Labor in Primipara: A Retrospective Cohort Study

**DOI:** 10.3390/jcm11123519

**Published:** 2022-06-19

**Authors:** Ning-Shiuan Ting, Dah-Ching Ding, Yu-Chi Wei

**Affiliations:** 1Department of Obstetrics and Gynecology, Hualien Tzu Chi Hospital, Buddhist Tzu Chi Foundation, Tzu Chi University, Hualien 970, Taiwan; 101311117@gms.tcu.edu.tw; 2Institute of Medical Sciences, College of Medicine, Tzu Chi University, Hualien 970, Taiwan

**Keywords:** prostaglandin, induction of labor, cervical ripening, primipara, slow release

## Abstract

This retrospective study aimed to compare the safety and efficacy of Prostin E2 and Propess for the induction of labor (IOL) in nulliparous women between January 2018 and October 2021. The inclusion criteria were nulliparous, singleton, >37 weeks’ gestation, cephalic presentation with an unfavorable cervix (Bishop score ≤ 6), no signs of labor, and use of one form of dinoprostone (Prostin E2 or Propess) for IOL. The cesarean section (C/S) rate and induction-to-birth interval were the main outcome measures. In total, 120 women were recruited. Sixty (50%) patients received Propess and 60 (50%) received repeated doses of Prostin E2. The Prostin E2 and Propess groups had similar patient characteristics, but the Bishop score was significantly higher in the Propess group than in the Prostin E2 group; therefore, multivariate analysis was conducted, and the Bishop score was not associated with the induction-to-birth interval. The C/S rate was not significantly different between the two groups, but the Propess group achieved a shorter induction-to-birth interval, a higher rate of vaginal delivery in 24 h, and a lower number of vaginal examinations than the Prostin E2 group. Propess was effective and safe in IOL and could be an option for cervical ripening in nulliparous pregnancy.

## 1. Introduction

Induction of labor (IOL) is an obstetric intervention defined as artificially initiating a labor course of uterine contraction and cervical change before spontaneous onset of labor, which results in vaginal delivery [1]. The incidence of IOL varies by region and has increased from 10% to 26% in the United States in the last 30 years [2]. In China, the labor induction rate is 18.4% in nullipara and 10.2% in multipara from 2015 to 2016 [3]. Traditional obstetrical theory considers IOL to lead to an increased cesarean section (C/S) rate [4]. Recent evidence from the ARRIVE trial demonstrated that IOL in low-risk nulliparous women at 39 weeks’ gestation reduced the rates of C/S (18.6% versus vs. 22.2%; *p* < 0.001) and hypertensive disorders (9.1% vs. 14.1%; *p* < 0.001) [5]. The results of the trial are also comparable with those of other meta-analyses and these findings led to a transformation of the concept of IOL and made IOL more acceptable in clinical practice [6,7,8].

Pre-induction cervical ripening is an important factor for successful labor induction. Cervical ripening techniques include pharmacological, mechanical, and nitric oxide donor techniques [9]. Pharmacological techniques include prostaglandins (PGs), such as misoprostol and dinoprostone. Misoprostol (Cytotec) is a synthetic PG E1 that was originally used for peptic ulcer treatment, but it has been used off-label in obstetrics for pre-induction cervical ripening for many years [10]. Although there is widespread off-label use for IOL, adverse effects, such as uterine hyperstimulation and further risk of uterine rupture, are of greater concern to clinicians [10]. Dinoprostone is a synthetic analog of PG E2 (PGE2) that has been used in IOL since the 1960s [11]. It is commercially available in several forms including tablets, gels, suppositories, and pessaries.

Prostin E2 tablets contain 3 mg of dinoprostone and have a similar rate of delivery within 24 h and mode of delivery compared with Prostin gel based on a previous study [12].

Prostin E2 gel (Prepidil) was the most commonly used form in the United Kingdom with a 3–5% failure rate, although it may precipitate uterine hyperstimulation and a non-reassuring fetal heart rate [13]. Thus, alternatives for Prostin gel, such as Propess (Prostin pessary), were sought because of this situation [14].

The dinoprostone vaginal pessary (Propess) contains 10 mg of dinoprostone and has the control-release (0.3 mg/h) characteristic of the single-dose application [15]. When uterine tachysystole or non-reassuring fetal heart rate occurs, the knitted polyester retrial system allows for the quick removal of the pessary. A vaginal pessary is considered easier to administer, less invasive, and cost-effective compared with an instant-release pessary [14].

The objective of this study was to compare the efficacy (C/S rate, induction-to-birth interval, and rate of vaginal delivery <24 h) and safety (neonatal outcomes) of two dinoprostone preparations, the Prostin E2 tablet (Pfizer S.A., Brussels, Belgium) and Propess (Ferring, Middlesex, UK), for IOL in primipara. We assumed that the efficacy of Propess would be better than that of Prostin E2 tablets in IOL.

## 2. Materials and Methods

### 2.1. Ethics Statements

The study was approved by the Research Ethics Committee of Hualien Tzu Chi Hospital (number: IRB111-045-B). The requirement for informed consent was waived because of the low risk to the patients in the study.

### 2.2. Study Design and Population

This single-institution retrospective cohort study was conducted at Hualien Tzu Chi Hospital, a tertiary medical center, from January 2018 to October 2021. The inclusion criteria were nulliparous pregnant women at >37 weeks’ gestation with an unfavorable cervix (Bishop score < 6) who used Prostin E2 or Propess for IOL, a singleton pregnancy, vertex presentation, no signs of labor, and a reassuring fetal heart rate pattern. The exclusion criteria were multiparous pregnancy, any contraindications for vaginal delivery, rupture of the membrane before starting IOL, and use of two forms of dinoprostone (Prostin E2 or Propess) in the first 24 h of the induction course. Gestational age was calculated using the Naegele rule. The study population included 60 cases each in the Propess and Prostin E2 groups (Figure 1).

### 2.3. Procedure of Induction of Labor

The pregnant woman was admitted to our delivery room and the fetal heart rate was monitored for 30 min. We performed the first vaginal examination during the insertion of Prostin E2 or Propess and the timing of further vaginal examination was based on uterine contractions and the patient’s symptoms. We performed artificial rupture of the membrane (AROM) when the Bishop score was >8 and used oxytocin for augmentation if uterine contraction was poor after AROM. The second dose of Prostin E2 was administered if the Bishop score was <8 at 4 h after the first dose. The process was stopped when the following situations occurred: rupture of the membranes, non-reassuring fetal heart rate, or uterine tachysystole.

### 2.4. Data Collection

Patients’ basic data such as age, body mass index (BMI), obstetrical history, initial Bishop score, time of delivery, mode of delivery, neonatal weight, and Apgar score were recorded from medical records.

### 2.5. Outcome Measures

The primary outcome was the rate of cesarean section. Secondary outcomes included the induction-to-birth interval and rate of vaginal delivery within 24 h.

### 2.6. Statistical Analyses

Normally distributed data are presented as mean ± standard deviation and the independent-samples *t*-test was used to compare differences between group means. The number (%) and differences between the two groups were compared using the chi-square test. Multivariate regression analysis was used to predict variables related to the results. Statistical significance was set at *p* < 0.05. All statistical analyses were conducted using SPSS, version 25 (IBM Corp., Armonk, NY, USA). We also used G*Power 3.1.9.2 to calculate the sample size needed. For the comparison of mean difference of main outcome (induction-to-birth interval) between Propess and Prostin E2 group, we set effect size of 0.52, α of 0.05, power (1 − β) of 0.80, Propess to Prostin E2 sample size ratio of 1, and two-sided test, then obtained the estimated sample size 120 (i.e., 60 per group).

## 3. Results

### 3.1. Patient Characteristics

In total, 133 nulliparous pregnant women were recruited for the study. Two women were excluded because of the simultaneous use of mechanical induction and another 11 women who had used Propess and added Prostin E2 for augmentation in the first 24 h were also excluded (Figure 1). Demographic and baseline characteristics of the patients are presented in Table 1. The rate of painless use (76.7% versus vs. 91.7%, *p* = 0.024) and Bishop score at admission (1.27 ± 1.64 vs. 2.18 ± 1.86; *p* = 0.005) were significantly higher in the Propess group than in the Prostin E2 group. Maternal age, gestational age, the BMI, and AROM rate were not significantly different between the two groups. Maternal request was the most common indication for induction in both groups (Prostin E2, 61.6%; Propess, 60.0%), followed by large for gestational age (Prostin E2, 13.3%; Propess, 16.6%) (Table 1). There was no statistical difference in the indication for induction between the groups (*p* = 0.147). The reasons for the maternal requests were late-pregnancy-associated discomfort and doctor suggestions (according to ARRIVE trial).

### 3.2. Efficacy of Propess and Prostin E2

The rate of C/S was not significantly different between the groups (Prostin E2, 10.0%; Propess, 3.3%; *p* = 0.272) (Table 2). The induction to vaginal delivery interval (24.47 ± 16.77 vs. 16.38 ± 8.87, *p* = 0.002) and rate of delivery within 24 h (53.3% vs. 78.3%, *p* = 0.004) were significantly shorter in the Propess group than in the Prostin E2 group. No significant difference was observed in the rate of oxytocin augmentation between the groups (Table 2).

### 3.3. Neonatal Outcome

Regarding neonatal outcomes, the birth body weight (Prostin E2, 3174.68 ± 315.52 g; Propess, 3109.58 ± 296.62 g; *p* = 0.247) and Apgar scores < 7 at 1 min (Prostin E2, 13.3%; Propess, 15.0%; *p* = 0.793) and 5 min (Prostin E2, 8.3%; Propess, 0%; *p* = 0.057) were not significantly different between the groups (Table 3). The Propess group showed a significantly higher Apgar score at 5 min than the Prostin E2 group (8.97 ± 0.32 vs. 8.75 ± 0.73, *p* = 0.037) (Table 3).

### 3.4. Stratified Analysis of the Factors Associated with an Induction-to-Birth Interval <24 h

As shown in Table 4, BMI was associated with a short interval (β [95% confidence interval (CI)]: 0.97 [0.35, 1.59]). Conversely, Prostin E2 (β [95% CI]: 8.90 [3.88, 13.91]) and painless labor (β [95% CI]: 8.23 [1.45, 15.02]) were associated with a longer interval. Regarding induction-to-birth intervals <24 h, Prostin E2 showed a decreasing odds ratio after adjustment (0.26, 95% CI: 0.10–0.67).

## 4. Discussion

This study found that the cesarean section rates were comparable in both groups. The induction-to-birth interval was significantly shorter and the proportion of vaginal deliveries within 24 h was higher in the Propess group than in the Prostin E2 group. The success rates were higher (90% and 96.7% in the Prostin and Propess groups, respectively) than those in many other previous studies, with a success rate ranging from 71% to 90.4% [16,17,18,19,20,21,22,23,24]. We assumed that the efficacy of Propess would be better than that of Prostin E2 tablets in IOL. This finding suggests that Propess is an effective induction method for primiparas.

The induction of labor should replicate spontaneous labor as closely as possible and cervical ripening is a key process that results from gradual exposure to PGE2 [25]. A review stated that Propess offers a sustained, steady, and controlled release of PG and presents theoretical advantages [26]. As for Prostin E2, the release was unpredictable and irregular because the doses were not often repeated every 4 to 6 h, especially at midnight or busy times [27].

A previous retrospective study that included 33 patients with controlled-release dinoprostone (10 mg) revealed that the cesarean section rate was 12% and vaginal delivery within 24 h occurred in 51.6% of patients, with a medium time to delivery of 17.5 h in nulliparous women [28]. Tseng et al., reported that Propess was used for IOL in full-term pregnant women [29]. They found that 81.5% of the pregnant women achieved successful vaginal delivery. Multiparity was the only independent predictor of successful vaginal delivery (multipara, 100% vs. nullipara, 74.5%; *p* = 0.0018). In our study, the cesarean section rate was lower (3.3%) in the Propess group than in the Prostin E2 group, and vaginal delivery within 24 h occurred in 78.3% of patients, with a median time to delivery of 16.38 h in the nulliparous group, similar to the previous aforementioned study. The successful vaginal delivery rate in the Propess group was 96.7%. This difference may be related to the removal times in the Tseng et al., study and our study (12 h vs. 24 h).

Abdelaziz et al., conducted a prospective randomized clinical trial that recruited 200 term pregnant women (both nullipara and multipara) for labor induction using Propess and Prostin E2 preparations (ratio = 1:1) [22]. The probability of successful vaginal delivery was higher in the Prostin E2 tablet group (proportional hazard, 1.366; 95% CI, 1.010–1.847; *p* = 0.043). Multiparity is also associated with a higher probability of successful vaginal deliveries. Nevertheless, the Bishop score was not associated with successful vaginal delivery. Our study findings also agree with the Abdelaziz et al., results in that the Bishop score was not associated with the induction-to-birth interval. Nevertheless, we found that the induction-to-birth interval was significantly shorter and the proportion of vaginal deliveries within 24 h was also higher in the Propess group than in the Prostin E2 group. This difference may be due to different induction protocols. Propess was removed after effective uterine contraction, but it was also removed when uterine tachysystole was noted.

The fetal outcome in our study was good in both groups, suggesting that dinoprostone-related products for the induction of labor are safe and feasible. Tseng et al., reported that 98.1% (*n* = 52/53) of women underwent IOL in the vaginal delivery group, and the infant’s Apgar score at 5 min was >8. In our study, the Propess group also showed a significantly higher Apgar score at 5 min than the Prostin group regardless of the indication of induction.

Nevertheless, some indications for IOL after using Propess may cause unfavorable fetal conditions. Kansu-Celik et al., used Propess for IOL in women with oligohydramnios and the results showed an increased incidence of fetal distress in the Propess group compared with the normal control group (20 vs. 4.7%) [30]. This high proportion resulted in a high cesarean section rate [30]. In our study and the Tseng et al., study [29], the proportion of IOL in pregnant women with oligohydramnios was low (8/120 and 2/65, respectively). Therefore, further large-scale studies are needed to evaluate the safety of IOL in oligohydramnios using Propess.

Many randomized controlled studies have focused on the comparison of Propess and Prostin gel [14,27,31]. These studies showed that there were no significant differences between the two groups in the induction-to-delivery interval. Compared with studies on Propess vs. Prostin gel, there are few comparison studies on Propess vs. Prostin tablets. A previous randomized control study recruited 83 patients who received PGE2 tablets (3 mg) or vaginal gel (1 mg/2 mg) at 6 hourly intervals until the cervix was suitable for amniotomy. The cesarean section rate was 33.7% and the induction-to-delivery interval in primiparous women was 36 h [32]. In our study, Prostin E2 tablets were administered at 4 hourly intervals until the Bishop score was >8; the cesarean section rate in the Prostin E2 tablets group was 10% and the induction-to-delivery interval was 24.47 h.

Concerning the different results between our study and the previous aforementioned study, we had a similar perspective on the Cochrane review for vaginal PGs (PGE2 and PG F2a) for the induction of labor at term [11]. There is a wide variation in the doses and costs of the various types of medications, even in the method of use. The use of PGs did not increase the need for cesarean section. There is insufficient evidence to compare the different types of medications and more research is needed on the effectiveness and safety of prostaglandins.

The strength of this study is that it is the first to compare Propess and Prostin E2 for IOL in term pregnancy in nulliparous women. A prior retrospective study showed that multiparous pregnant women were the only independent predictors of successful induction after a dinoprostone slow-release vaginal insert [29]. Moreover, this study presented real-world data that prescribing Prostin E2 regularly might be difficult in situations of medical staff and resource shortages.

The limitations of our study are its retrospective design, the small sample size, and the lack of a standardized protocol, especially the timing of Prostin E2 tablet insertion, which might have underestimated the efficacy of the Prostin E2 tablet. Furthermore, the cost-effectiveness and the patient satisfaction of the two groups require further evaluation.

## 5. Conclusions

In conclusion, Propess was found to be effective and safe in IOL and could be an option for cervical ripening in primiparas. Nevertheless, a large prospective cohort study is needed to evaluate the cost-effectiveness and patient satisfaction of Propess.

## Figures and Tables

**Figure 1 jcm-11-03519-f001:**
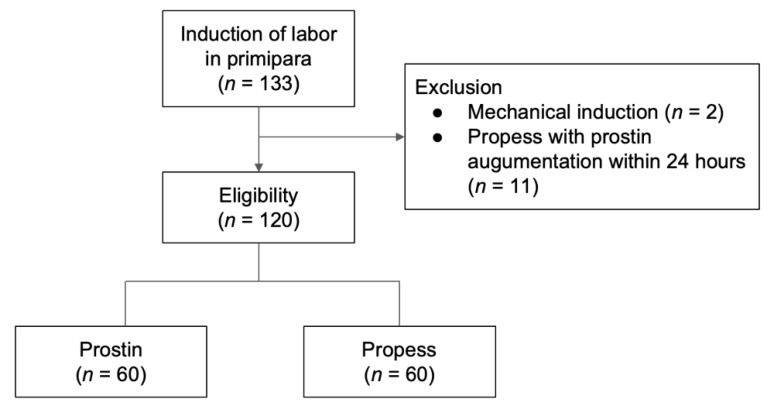
Study flowchart.

**Table 1 jcm-11-03519-t001:** Demographic and baseline characteristics.

	Prostin E2 Tablet (*n* = 60)	Propess (*n* = 60)	Total (*n* = 120)	*p*-Value
Age (years), mean ± SD	28.5 ± 6.68	28.23 ± 5.1	28.37 ± 5.92	0.806
BMI (kg/m^2^), mean ± SD	28.31 ± 3.67	28.29 ± 4.65	28.3 ± 4.17	0.979
Gestational age (weeks), mean ± SD	39.2 ± 0.88	39.07 ± 0.73	39.14 ± 0.81	0.405
Gestational age (weeks), median (Q1, Q3)	39.29 (38.61, 39.86)	39.14 (38.57, 39.71)	39.21 (38.57, 39.71)	0.273
Painless use, *n* (%)	46 (76.7%)	55 (91.7%)	101 (84.2%)	0.024 *
Membrane status during hospitalization				
AROM, *n* (%)	36 (60.0%)	31 (51.7%)	67 (55.8%)	0.358
ROM, *n* (%)	23 (38.3%)	29 (48.3%)	52 (43.3%)	0.269
Bishop score at admission, mean ± SD	1.27 ± 1.64	2.18 ± 1.86	1.73 ± 1.81	0.005 *
Bishop score at admission				0.024 *
0–2, *n* (%)	47 (78.3%)	33 (55.0%)	80 (66.7%)	
3–4, *n* (%)	9 (15.0%)	17 (28.3%)	26 (21.7%)	
5–6, *n* (%)	4 (6.7%)	10 (16.7%)	14 (11.7%)	
Indications for induction, *n* (%)				0.147
Maternal request	37 (61.6%)	36 (60.0%)	73 (60.8%)	
Large for gestational age	8 (13.3%)	10 (16.6%)	18 (15.0%)	
Late-term pregnancy	6 (10.0%)	0	6 (5.0%)	
Pre-eclampsia	2 (3.3%)	1 (1.6%)	3 (2.5%)	
Pregnancy-induced hypertension	2 (3.3%)	2 (3.3%)	4 (3.3%)	
Oligohydramnios	2 (3.3%)	6 (10.0%)	8 (6.6%)	
Fetal growth restriction	1 (1.6%)	0	1 (0.8%)	
Gestational diabetes mellitus	1 (1.6%)	4 (6.6%)	5 (4.1%)	
Severe PUPPP	1 (1.6%)	0	1 (0.8%)	
Nonreactive NST result	0	1 (1.6%)	1 (0.8%)	

* *p* < 0.05. BMI—body mass index, SD—standard deviation, Q—quarter, AROM—artificial rupture of the membranes, PUPPP—pruritic urticarial papules and plaques of pregnancy, NST—nonstress test.

**Table 2 jcm-11-03519-t002:** Summary of the efficacy of Propess and Prostin E2.

	Prostin E2 Tablet (*n* = 60)	Propess (*n* = 60)	Total (*n* = 120)	*p*-Value
Mode of birth				0.272
NSD + VED, *n* (%)	54 (90.0%)	58 (96.7%)	112 (93.3%)	
C/S, *n* (%)	6 (10.0%)	2 (3.3%)	8 (6.7%)	
Induction-to-birth interval				
NSD + VED (hours), mean ± SD	24.47 ± 16.77	16.38 ± 8.87	20.28 ± 13.83	0.002 *
C/S (hours), mean ± SD	50.84 ± 38.30	42.78 ± 1.15	48.82 ± 32.59	0.787
BS increases by ≥4 after 12 h, *n* (%)	36 (60.0%)	48 (80.0%)	84 (70.0%)	0.017 *
BS increases by ≥4 after 24 h, *n* (%)	48 (82.8%)	59 (98.3%)	107 (90.7%)	0.004 *
Induction-to-birth interval <24 h, *n* (%)	32 (53.3%)	47 (78.3%)	79 (65.8%)	0.004 *
BS ≤6 after 24 h, *n* (%)	15 (25.9%)	2 (3.3%)	17 (14.4%)	<0.001 *
Oxytocin augmentation, *n* (%)	20 (33.3%)	16 (26.7%)	36 (30.0%)	0.426
Vaginal examination frequency (mean ± SD)	12.57 ± 6.69	7.98 ± 2.88	10.28 ± 5.62	<0.001 *

* *p* < 0.05. BS—Bishop score, NSD—normal spontaneous delivery, C/S—cesarean section, VED—vacuum extraction delivery, SD—standard deviation.

**Table 3 jcm-11-03519-t003:** Neonatal outcomes.

	Prostin E2 Tablet (*n* = 60)	Propess (*n* = 60)	Total (*n* = 120)	*p*-Value
BBW, mean ± SD	3174.68 ± 315.52	3109.58 ± 296.62	3142.13 ± 306.67	0.247
Apgar score at 1 min, mean ± SD	7.88 ± 0.98	7.95 ± 0.7	7.92 ± 0.85	0.668
Apgar score ≤ 7 at 1 min, *n* (%)	8 (13.3%)	9 (15.0%)	17 (14.2%)	0.793
Apgar score at 5 min, mean ± SD	8.75 ± 0.73	8.97 ± 0.32	8.86 ± 0.57	0.037 *
Apgar score ≤ 7 at 5 min, *n* (%)	5 (8.3%)	0 (0.0%)	5 (4.2%)	0.057

* *p* < 0.05. BBW—baby birth weight, SD—standard deviation.

**Table 4 jcm-11-03519-t004:** Factors associated with the induction-to-birth interval and induction-to-birth interval <24 h.

Item	Induction-to-Birth Interval		Induction-to-Birth Interval <24 h	
	β (95% CI)	*p*-Value	OR (95% CI)	*p*-Value
Age	0.31 (−0.11, 0.72)	0.15	0.94 (0.87, 1.01)	0.113
BMI	0.97 (0.35, 1.59)	0.003 *	0.91 (0.81, 1.02)	0.099
Gestational age	1.32 (−1.77, 4.40)	0.399	1.08 (0.61, 1.91)	0.791
Group (Prostin E2 tablet vs. Propess)	8.90 (3.88, 13.91)	0.001 *	0.26 (0.10, 0.67)	0.005 *
Painless use (yes vs. no)	8.23 (1.45, 15.02)	0.018 *	0.36 (0.08, 1.55)	0.171
AROM (yes vs. no)	0.06 (−4.83, 4.94)	0.982	1.1 (0.43, 2.79)	0.84
Bishop score at admission	−0.28 (−1.63, 1.08)	0.685	1.18 (0.9, 1.55)	0.222

* *p* < 0.05. CI—confidence interval; OR—odds ratio; AROM—artificial rupture of membranes; vs.—versus; BMI—body mass index.

## Data Availability

All relevant data are reported in the article.

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
