# Peer review of "Comparison of the Dinoprostone Vaginal Insert and Dinoprostone Tablet for the Induction of Labor in Primipara: A Retrospective Cohort Study"

_jcm, 2022, doi:10.3390/jcm11123519_

Round 1
Reviewer 1 Report
The topic of the investigation is interesting and in general it is well written but I think there are some points to clarify: In the introduction you expose the data about labor induction rates in the US and I think it would be interesting to also include the induction rate in your environment.I believe that it is necessary to clarify the objective of the study.
In the results section, you comment that maternal request is the most common indication for labor induction. I think it would be interesting to clarify under what circumstances women can request labor induction.
Author Response
The topic of the investigation is interesting and in general it is well written but I think there are some points to clarify: In the introduction you expose the data about labor induction rates in the US and I think it would be interesting to also include the induction rate in your environment.
Response: We don’t have data in Taiwan. We added the data on China in the introduction section.
I believe that it is necessary to clarify the objective of the study.
Response: We added the objective of the study (introduction).
In the results section, you comment that maternal request is the most common indication for labor induction. I think it would be interesting to clarify under what circumstances women can request labor induction.
Response: We added the reason for the maternal request in the results section (section 3.1).
Reviewer 2 Report
Table 3 says: BBW: baby birth wight
Should say: BBW: baby birth weight
Table 4. Authors shoul write the meaning of BMI at the botton of table.
Author Response
Table 3 says: BBW: baby birth wight, Should say: BBW: baby birth weight
Response: We changed “wight” to “weight” in Table 3.
Table 4. Authors should write the meaning of BMI at the botton of table.
Response: We added the full spelling of BMI at the bottom of the table.
Reviewer 3 Report
The paper is well written, but still need some english language editing. The weak point is the endpoint definition, with the C/S as the basic endpoint that is not different between the groups. Then you need to calculate the power to be allowed to do the "nitty gritty" statistical evaluaton. We need to define the statistical power needed for the rather complicated models you suggest for the assessment. Please look at that and re-define the statistical calculations. The design of the study and design of the paper are fair and relevant for a clinical study as this.
Author Response
The paper is well written, but still need some english language editing.
Response: We sent the manuscript for English editing (Editage company).
The weak point is the endpoint definition, with the C/S as the basic endpoint that is not different between the groups.
Response: Indeed, both medications were equally successful for induction of vaginal delivery. Nevertheless, the second endpoints including induction-to-birth interval and rate of vaginal delivery within 24 hours were all significantly different between the two groups.
Then you need to calculate the power to be allowed to do the "nitty gritty" statistical evaluaton. We need to define the statistical power needed for the rather complicated models you suggest for the assessment. Please look at that and re-define the statistical calculations.
Response: Thanks for your kind reminding. We had specified the calculation of sample size needed to achieve the goal of power (0.80) in Materials and Methods section. “We used G*Power 3.1.9.2 to calculate the sample size needed. For the comparison of mean difference of main outcome (induction-to-birth interval) between Propess and Prostin E2 group, we set effect size of 0.52, α of 0.05, power(1-β) of 0.80, Propess to Prostin E2 sample size ratio of 1, and two-sided test then got the estimated sample size 120 (i.e. 60 per group).”
The design of the study and design of the paper are fair and relevant for a clinical study as this.
Response: Thanks for your comments.
Round 2
Reviewer 3 Report
Relevant comments are addressed now